# Phytochemicals, Organic Acid, and Vitamins in Red Rhapsody Strawberry—Content and Storage Stability

**DOI:** 10.3390/foods14030379

**Published:** 2025-01-24

**Authors:** Hung Trieu Hong, Julius Rami, Michael Rychlik, Tim J. O’Hare, Michael E. Netzel

**Affiliations:** 1Queensland Alliance for Agriculture and Food Innovation, The University of Queensland, Coopers Plains, Brisbane, QLD 4108, Australia; t.ohare@uq.edu.au; 2School of Agriculture and Food Sustainability, The University of Queensland, St Lucia, Brisbane, QLD 4072, Australia; 3Department of Analytical Food Chemistry, Technical University of Munich, 85354 Freising, Germany; julius.rami@gmail.com (J.R.); michael.rychlik@tum.de (M.R.)

**Keywords:** strawberry, folates, vitamin C, anthocyanins, ellagic acid, organic acids, storage stability, nutrition

## Abstract

Strawberries are highly perishable fruits harvested at full ripeness, and their nutritional quality together with their phytochemical composition can be significantly affected by storage duration and temperature. This study investigated the changes in key bioactive compounds, including folate, vitamin C, anthocyanins, quercetin-3-glucoside, ellagic acid, and organic acids, in “Red Rhapsody” strawberries stored at two typical household temperatures (4 °C and 23 °C). While storage duration and temperature did not have a significant impact (*p* > 0.05) on folate content, significant changes in other phytochemicals were observed. The total anthocyanin content increased significantly (*p* < 0.05), from 30.0 mg/100 g fresh weight (FW) at Day 0 to 84.4 mg/100 g FW at Day 7 at 23 °C, a 2.8-fold increase. Conversely, the vitamin C content was significantly reduced (*p* < 0.05), from 54.1 mg/100 g FW at Day 0 to 28.4 mg/100 g FW at Day 7 at 23 °C, while it remained stable at 4 °C. Additionally, the concentrations of quercetin-3-glucoside, ellagic acid, and organic acids underwent significant changes during the storage period. The total folate content fluctuated between 73.2 and 81.6 μg/100 g FW at both temperatures. These results suggest that storage temperature and duration influence the individual phytochemicals and nutrients of strawberries differently, with potential implications for their nutritional value and bioactive compound content.

## 1. Introduction

Strawberries (*Fragaria ananassa*) are widely consumed for their taste and health benefits. The global market value of strawberries was approximately USD 20.22 billion in 2023 and is expected to reach USD 32.35 billion by 2032 [1]. Strawberries are an important dietary source of natural bioactive compounds, such as anthocyanins, quercetin, ascorbic acid, ellagic acid, and folates, which are associated with various health benefits. Strawberries are particularly recognized as an anthocyanin-rich fruit, with total anthocyanin contents ranging from 24.8 to 101.0 mg/100 g fresh weight (FW) [2]. Anthocyanins in strawberries have been associated with various health-promoting properties, including enhancing heart health [3], possessing antioxidant functions [2], preventing high blood pressure [4], and having positive effects on degenerative diseases and cardiovascular disorders [5]. Strawberries are also a valuable natural source of folates, with folate content ranging from 59 to 153 µg/100 g FW [6,7]. Folates, a group of B vitamins, play crucial roles in various metabolic pathways. Tetrahydrofolate (H4folate) polyglutamates in particular are involved in one-carbon metabolism, which supports nucleotide biosynthesis, the re-methylation of homocysteine to methionine, amino acid production in mitochondria, and DNA replication and repair in the nucleus [8,9,10]. As the human body cannot synthesize folates, a deficiency can result in neural tube defects in infants [11], as well as decreased methionine levels and increased homocysteine levels, both of which are linked to cardiovascular diseases [12,13] and Alzheimer’s disease [14].

Ellagic acid, quercetin, and vitamin C (ascorbic acid) are other important bioactive compounds found in strawberries [15,16]. Ellagic acid is known for its antioxidant properties [17], and its potential in preventing cancer and cardiovascular diseases [18]. Ascorbic acid is essential for collagen biosynthesis [19], acts as an antioxidant [20], and has protective effects against Alzheimer’s disease [21] and in cancer treatment [22]. Quercetin is widely recognized for its anti-inflammatory, antioxidant, antiviral, and anticancer properties, as well as its ability to ease cardiovascular diseases [23].

The primary bioactive compounds in strawberries, such as anthocyanins, folates, ascorbic acid, and organic acids, are well known to be highly sensitive to environmental factors, such as temperature, light exposure, and storage [24,25,26]. Elevated temperatures and extended storage times can result in the degradation of these previously reported health-benefitting compounds. For instance, anthocyanins are particularly unstable under prolonged exposure to heat and light [27]. Similarly, ascorbic acid content in guava juice decreased significantly when stored at higher temperatures [28]. Folates also undergo significant degradation during storage, particularly at higher temperatures or prolonged storage periods [24]. These changes in the phytochemical and micronutrient composition can potentially lead to a loss of the nutritional value of strawberries and their health-promoting potential. Understanding how common storage temperatures affect the stability of these compounds is critical for optimizing postharvest handling and prolonging the shelf life of strawberries without compromising their nutritional quality.

Strawberries are typically harvested at a ripe eating stage and stored under optimal conditions (0 °C) to extend their postharvest lives, as they are highly perishable and delicate fruits [29]. However, strawberries are often stored at refrigerated (4 °C) or room temperature (23 °C) for a few days in retail settings and at home before consumption. Extended cold storage can cause phytochemical changes, as recent studies have reported a significant increase in anthocyanin content (a 1.5-fold increase) when strawberries are stored at 2 °C [30]. How domestic storage conditions affect key phytochemicals and micronutrients, such as anthocyanins, ellagic acid, quercetin-3-glucoside, ascorbic acid and folates, is not well documented. The aim of this study was to evaluate the effect of two common household storage temperatures (4 °C and 23 °C) on key phytochemicals and micronutrients (anthocyanins, ellagic acid, quercetin-3-glucoside, ascorbic acid and folate) in “Red Rhapsody” strawberries, a commercially significant cultivar in Australia, over a 14-day storage period.

## 2. Material and Methods

### 2.1. Plant Material

Fresh “Red Rhapsody” strawberries (7 kg) were obtained from a commercial farm in Brisbane, Queensland, Australia (Figure 1). Samples were transported (1 h) to the laboratory at the Health and Food Sciences Precinct, Coopers Plains. Samples were selectively divided into two equal batches (based on the size of fruits, the observation of pigment intensity, and red pigment coverage to make sure all fruits used for the current trial are identical) for storage at 4 °C and 23 °C, respectively. For the control group of samples (0 day), five strawberries were then collected and rapidly frozen using liquid nitrogen to halt any further metabolic processes. All samples were stored at −35 °C until analysis. The remaining samples were placed in commercial plastic punnets, which were loosely sealed in plastic barrier bags and stored in two separate batches: batch 1 at 4 °C and batch 2 at 23 °C, with a humidity level of 90% ± 3%. The humidity level was set based on the moisture content of the strawberry fruit and previous publications [27]. The humidity was monitored using a digital hygrometer thermometer (ThermoPro TP55, Guangdong, China).

Five strawberries were removed at 1, 7, and 14 days of storage at 4 °C and at 1, 3, and 7 days of storage at 23 °C. These samples were immediately frozen at −35 °C for later analysis. The frozen strawberries were freeze-dried at −50 °C (CSK Climateck, CSK Scientific, Brisbane, QLD, Australia), then ground into a powder using a Waring Laboratory Blender (Australian Scientific, Kotara, NSW, Australia). Powdered samples were stored at −80 °C for later analysis of individual and total anthocyanins, moisture, ascorbic acid, organic acids, folates, ellagic acid, and quercetin-3-glucoside.

### 2.2. Chemicals

Cyanidin-3-glucoside (Cy3G), pelargonidin-3-glucoside (Pg3G), quercetin-3-glucoside, citric acid, succinic acid, ellagic acid, and ascorbic acid (AA) were sourced from Sigma–Aldrich (Sydney, NSW, Australia) and Extrasynthese (Genay, France). Both the unlabeled and isotope-labeled internal standards were sourced from Professor Michael Rychlik (Analytical Food Chemistry, Technical University of Munich, Freising, Germany). All other chemicals were acquired from Merck (Darmstadt, Germany) or Sigma–Aldrich. Milli-Q water (Millipore Australia Pty Ltd., Kilsyth, VIC, Australia) was used throughout the study unless otherwise stated.

### 2.3. Methods

#### 2.3.1. Folate Extraction and Determination

A stable isotope dilution assay (SIDA) method was utilized to analyze folate concentrations in strawberry samples stored at different temperatures and durations using a UHPLC-PDA-MS/MS system [7] with some modifications. Briefly, 50–100 mg of freeze-dried strawberry was suspended in 10 mL of MES-buffer (pH 5). A magnetic stirrer bar was placed in the mixtures and left to equilibrate on a magnetic stirrer (IKA C-MAG HS 10, IKA, Breisgau, Germany) for 15 min. The ^13^C-labeled internal standard (IS), including [^13^C5]-PteGlu (^13^C-THF), [^13^C_5_]-H_4_folate (^13^C-FA), [^13^C_5_]-5-CH_3_-H_4_folate (^13^C-5mTHF), and [^13^C_5_]-5-CHO-H_4_folate (^13^C-5fTHF), were dissolved in MES-buffer to achieve a final concentration of 60–70 µg/L. These stock solutions were added to the sample mixtures to achieve a final concentration of ~1 µg/L for ^13^C-5mTHF, 0.05 µg/L for ^13^C-5fTHF and ^13^C-THF, and 0.015 µg/L for ^13^C-FA in the final extraction solutions (2 mL). The resulting mixtures after adding IS were equilibrated for a further 15 min.

The samples were cooked at a boiling temperature of 100 °C for 10 min (Buechi Heating Bath B-491, Buechi AG, Aargau, Switzerland) and cooled down in an ice bath. After reaching room temperature, chicken pancreas (2 mL) and rat serum (300 μL) were added to each sample. The samples were then incubated in a water bath (Thermoline Scientific, Wetherill Park, NSW, Australia) at 37 °C overnight. The samples were subsequently reheated again at 100 °C for 10 min and then cooled in an ice bath. The suspension was transferred to a 50 mL centrifuge tube, diluted with 10 mL acetonitrile and centrifuged at 3900 rpm (Eppendorf centrifuge 5810, Eppendorf AG, Hamburg, Germany) for 20 min. The supernatant was decanted into a fresh 50 mL centrifuge tube and the pellet discarded.

The supernatant (approximate 20 mL) was purified by solid-phase extraction (SPE, Anion-exchange SAX-cartridges, Phenomenex Inc., Lane Cove, NSW, Australia). Folates were eluted from the SPE by using 2 mL buffer including 5% sodium chloride, 100 mM sodium acetate, 0.1 g/L dithiothreitol and 1 g/L ascorbic acid. The final eluate was filtered through a 0.20 µm hydrophilic PTFE membrane for folate analysis by a Shimadzu UHPLC-ESI-MS/MS following the previously instrumental method [6]. Folates were identified and then quantified following the previous study of Striegel, Chebib [6].

#### 2.3.2. Ascorbic Acid Extraction and Determination

Ascorbic acid (AA) in strawberry samples was extracted following the previous report of Campos, Ribeiro [31], with some modifications. Briefly, powdered samples (from 50 to 100 mg) were homogenized with 15 mL of a buffer containing 8% acetic acid, 3% meta-phosphoric acid, 1 mM and ethylenediamine tetraacetic acid (EDTA). Dehydroascorbic acid (DH-AA) in the extracts/samples was also converted to AA following the previous method [32].

AA was determined using a Shimadzu Nexera X2 UHPLC system coupled with a Shimadzu MS-8045 TQMS, triple quadrupole mass spectrometer (Shimadzu, Kyoto, Japan), controlled by Lab Solutions software (Version 5.91). The column oven (CTO-20AC) was maintained at 25 °C. The electrospray ionization (ESI) source was operated in the negative ion mode to ionize AA. Multiple reaction monitoring (MRM) transitions for AA (Table 1) were *m*/*z* 175.1 → 115.2 at 14 eV(quantifier) and *m*/*z* 175.1 → 87.1 at 20 eV(qualifier). The ESI source included a nebulizer gas flow of 3 L/min, a desolvation line (DL) temperature of 250 °C, a drying gas flow of 10 L/min, and a heat block temperature of 400 °C. Data collection was carried out using Shimadzu LabSolutions Insight software for LC-MS (Version 3.2).

Chromatographic separation of AA and other organic acids was carried out on an UPLC column (BEH AcclaimTM C30, 250 × 2.1 mm i.d., 3.0 μm particle size; Thermo Scientific, Waltham, MA, USA), with the column temperature being kept at 25 °C. Two mobile phases were used to elute AA and organic acids, including mobile phase A (MQ-water) and mobile phase B (acetonitrile). The column flow rate was maintained at 0.2 mL/min throughout the analysis. The elution program started with 100% mobile phase A for 5.8 min, followed by a linear increase to 95% mobile phase B for over 6.5 min, purging for 1 min, conditioning for 1 min, and re-equilibration for 3.5 min.

The linearity was obtained by using concentrations of seven-external standard (AA, citric acid and succinic acid) concentrations over the range 1–40 µg/mL plotted against their peak-area measurements by using LC-MS. The initial AA concentration was subtracted by the total AA (TAA) obtained after a reduction in AA extract with dithiothreitol (DTT) to account for DH-AA contents in all samples.

#### 2.3.3. Organic Acid Extraction and Determination

Organic acids were extracted following the method previously described by Hernandez, Lobo [33,34] with some modifications. In brief, freeze-dried powder strawberry (0.1 g) was extracted with 30 mL of MQ-water using an ultra-sonication bath at room temperature (RT) for 15 min and then a shaker (RP1812, Paton Scientific, Victor Harbor, SA, Australia) at 200 rpm at RT. Samples were then centrifuged (Eppendorf Centrifuge 5804) at 3900 rpm for 10 min at RT. Supernatant was collected, while the sample pellet was re-extracted using the same procedure outlined above. Supernatants were combined and filtered through a 0.20 µm hydrophilic PTFE membrane for organic acid analysis using a Shimadzu UHPLC-ESI-MS/MS.

Citric acid and succinic acid were determined by using the above Shimadzu LC-MS/MS system. The electrospray ionization (ESI) source was employed in the negative mode. The MRM transitions were used at optimal collision energies for succinic acid and citric acid (Table 1).

#### 2.3.4. Ellagic Acid Anthocyanins and Quercetin-3-Glucoside Extraction and Determination

Ellagic acid, anthocyanins, and quercetin-3-glucoside extraction was extracted following the previous method of Hong, Netzel [25], with some modifications. Briefly, approximately 0.2 g of freeze-dried powder was homogenized with 3 mL of cold extraction solution (80% methanol in water, 0.1 M HCl) at 4 °C in a 15 mL Falcon tube. The tubes were sonicated at 4 °C, for 10 min. The mixture was then shaken on a horizontal reciprocating shaker RP 1812 (Paton Scientific, Victor Harbor, SA, Australia) at 250 rpm, 4 °C for 10 min under dim light. The tubes were centrifuged at 4000 rpm, 4 °C for 10 min (Eppendorf Centrifuge 5804). The supernatants were collected, and the pellets were extracted again using the same procedure as outlined above. Supernatants were combined and filtered through a 0.20 µm hydrophilic PTFE membrane for chemical analysis. The extraction was carried out in triplicate (n = 3).

#### 2.3.5. Moisture Content

The moisture contents of strawberry samples were determined in five replicates following the AOAC method 934.01 (AOAC, 1990) described previously [27].

### 2.4. Chemical Determination

Anthocyanins in strawberry samples were identified by LC-DAD-MS following the method previously reported Hong, Netzel [35]. Other compounds were detected by spiking external standards of AA, ellagic acid, citric acid, succinic acid, quercetin-3-glucoside, and folates to determine elution times and to confirm maximum absorbances on a DAD detector, as well as molecular masses and fragment patterns on the mass spectrometer (TQMS, MS-8045) and (MRM) transitions (Table 1).

### 2.5. Statistical Analysis

The obtained data from the current study were analyzed using a one-way analysis of variance (ANOVA) for each temperature performed by Minitab 17 software for Windows (Minitab Inc, State College, PA, USA). A Tukey’s test was employed to evaluate the results of the current study. Statistical differences between means were assessed using the least significant difference (LSD) method, with a significance level set at *p* < 0.05.

## 3. Results and Discussion

### 3.1. The Effect of Storage Durations on Folate Content

The total folate content in “Red Rhapsody” strawberries ranged from 74.3 to 81.6 μg/100 g fresh weight (FW). This range is consistent with previous reports on folate content in Australian strawberry cultivars, which ranged from similar values [6,7], and is notably higher than the 4.0–28.0 μg/100 g FW reported in other studies using different methods [36]. The predominant form of folate in strawberries was 5-methyltetrahydrofolate (5mTHF), which accounted for over 90% of the total folate content, with 5mTHF measured at approximately 74.8 μg/100 g FW. Other folate vitamers included tetrahydrofolate (THF), which was present at 2.4 μg/100 g FW, and 5-formyltetrahydrofolate (5fTHF) at 10 μg/100 g FW (Figure 2). Smaller amounts of folic acid (FA) and 10-formyl-folic acid (10fFA) were detected, with concentrations of 0.3 μg/100 g FW for each.

Storage conditions, including temperature and duration, did not significantly affect the total folate content in strawberries. At room temperature (23 °C), folate levels showed a slight, non-significant decrease over the 7-day storage period. The initial folate content was 78.7 μg/100 g FW on day 0 (D0), which decreased slightly to 73.2 μg/100 g FW on Day 3 (D3) and to 73.7 μg/100 g FW by Day 7 (D7). At lower temperatures (4 °C), even with storage durations extended up to two weeks, the folate content remained relatively stable at around 73.8 μg/100 g FW (Figure 2). The major folate component, 5mTHF, remained unchanged throughout the storage period regardless of temperature.

However, some changes were observed in other folate vitamers during storage. THF and FA concentrations increased significantly (*p* < 0.05) during the 7-day storage period at 23 °C. THF, in particular, increased from 2.4 μg/100 g FW at D0 to 3.4 μg/100 g FW at D7, while FA increased slightly from 0.2 to 0.5 μg/100 g FW. Notably, THF was the only folate form that showed a significant decrease at 4 °C and a significant increase at 23 °C, suggesting a potential temperature-dependent conversion or degradation process.

These findings highlight the fact that, despite minor fluctuations in individual folate components, the total folate content in strawberries is generally stable across the storage conditions. The increase in THF and FA content during storage at 23 °C suggests the possibility of conversions or interconversions among folate vitamers under high temperatures [37,38], which may be of interest for subsequent studies investigating folate metabolism in postharvest fruits.

### 3.2. The Effect of Storage Durations on Anthocyanin Content

Storage duration and temperature did not alter the anthocyanin profile in “Red Rhapsody” strawberries. Four anthocyanin components were identified: pelargonidin-3-glucoside (Pg-3-G), pelargonidin-3-rutinoside (Pg-3-R), cyanidin-3-glucoside (Cy-3-G), and pelargonidin-3-malonylglucoside (Pg-3-MG). Pg-3-G was the predominant anthocyanin, followed by Pg-3-R, with Cy-3-G and Pg-3-MG present in all samples regardless of storage period or temperature. The anthocyanin profile observed in this study was consistent with previous reports [2].

Although the anthocyanin components remained unchanged, significant increases in total anthocyanin content were observed with increasing storage durations at both temperatures (*p* < 0.05). At 23 °C, the total anthocyanin content was approximately 30 mg/100 g FW on D0. This value increased significantly (*p* < 0.05) to 51.2 mg/100 g FW on D1 and continued to rise to 76.9 mg/100 g FW on D3, followed by 84.4 mg/100 g FW on D7. At 4 °C, the increase in anthocyanin content was slower, with a significant (*p* < 0.05) rise observed only after 7 days of storage, reaching 41.1 mg/100 g FW on D7 and 44.9 mg/100 g FW on D14 (Figure 3A). The continued accumulation of anthocyanins throughout the storage period suggests that the enzymes involved in anthocyanin synthesis remained active in strawberry fruit tissues at both temperatures. These findings align with previous studies showing ongoing anthocyanin accumulation during the storage of strawberry [39], purple sweetcorn [27], and five primocane raspberry genotypes [40].

Similarly to TAC, the concentrations of individual anthocyanin components increased over the storage period, with a faster accumulation rate at 23 °C compared to 4 °C (Figure 3C,D). Among the components, Pg-3-R accumulated at the fastest rate. At 23 °C, its concentration on D7 was approximately 3.6 times higher than on D0 (Figure 3B). At 4 °C, Pg-3-R showed a two-fold increase by D14. In contrast, Cy-3-G exhibited the slowest accumulation rate at 23 °C, with a 1.8-fold increase on D7 compared to its concentrationont D0, while Pg-3-G had the slowest increase at 4 °C, showing only a 1.4-fold increase by D14 (Figure 3C,D). These results indicate that individual anthocyanin components accumulate at different rates depending on the storage temperature and duration.

### 3.3. The Effect of Storage Durations on Quercetin-3-Glucoside and Free Ellagic Acid Content

The increase in the concentration of free ellagic acid during the storage of strawberry fruits at both temperatures indicates that storage duration significantly affects free ellagic acid content (Figure 4). The free ellagic acid content was approximately 1.8 mg/100 g FW at D0, a value consistent with previous studies about free ellagic acid content in strawberries [15]. This concentration increased significantly to 2.2 at D1, 2.4 at D3, and 4.2 mg/100 g FW on D7 at 23 °C. At 4 °C, however, only a significant increase in the concentration of free ellagic acid was observed at D14, reaching about 3.0 mg/100 g FW. To the best of our knowledge, this is the first report documenting an increase in free ellagic acid in strawberry fruits during storage at both temperatures. The reason for this increase in free ellagic acid is currently unclear. However, it is noteworthy that the total ellagic acid content in strawberries is more than 10 times higher than the free ellagic acid content [15]. Therefore, it is possible that storage conditions led to the conversion of other forms of ellagic acid into free ellagic acid.

In contrast to free ellagic acid, the concentration of quercetin-3-glucoside (Quer-3-G) showed differing trends depending on storage temperature (Figure 4). While the concentration of Quer-3-G did not change significantly at 23 °C, a significant decrease was observed at 4 °C. Specifically, the concentration of Quer-3-G was 2.9 mg/100 g FW at D0 (consistent with previous findings by da Silva Pinto, Lajolo [15]), but it decreased to 2.0 mg/100 FW on D14. This decrease may be attributed to the continued activity of enzymes responsible for the biosynthesis of anthocyanins from quercetin-3-glucoside at 4 °C, which could lead to the utilization of Quer-3-G in the production of anthocyanins. However, the biosynthesis of quercetin-3-glucoside appears to be slower or inactive at this lower temperature.

The moisture content of strawberry fruit at both storage temperatures was not statistically significant (*p* > 0.05). These results suggest that the storage conditions effectively maintained the moisture content in the strawberries. Given the stability of moisture content, the observed changes in phytochemicals, organic acids, and vitamins in the strawberries reflect the physiological processes occurring in the fruit during storage rather than being influenced by moisture changes.

### 3.4. The Effect of Storage Durations on Ascorbic Acid Content

Storage duration had significant effects on the free ascorbic acid (AA), total ascorbic acid (TAA), and dehydroascorbic acid (DH-AA) content in strawberry fruit at both storage temperatures. Free AA concentration remained stable over the two-week storage period at 4 °C. However, at 23 °C, the concentration of free AA decreased significantly from 54.1 mg/100 g FW at D0 to 37.3 mg/100 g FW on D3, and it further declined to 28.4 mg/100 g FW on D7 (Figure 5A).

TAA also decreased at a faster rate at 23 °C than at 4 °C. In fresh strawberry fruit, TAA was measured at 63.3 mg/100 g FW, but it underwent a significant decline to 50.2 mg/100 g FW by D3 and 39.1 mg/100 g FW by D7 at 23 °C. In contrast, at 4 °C, only a significant reduction in TAA was observed after 14 days of storage, reaching 51.8 mg/100 g FW (Figure 5B). These findings are consistent with previous fruit studies that have reported a reduction in AA content during the storage of various fruits, such as oranges and pineapples stored at 10 °C and 20 °C [41], as well as strawberry fruit [42] and pears [43]. The degradation of AA during storage may help protect folates and other unstable compounds in strawberry fruit.

Figure 5C further supports the significant decrease in free AA and TAA content (Figure 5A,B). The percentage of DH-AA in strawberry fruit increased significantly (*p* < 0.05) from 14.6% at D0 to 17.9% on D1 and 27.4% on D7 at 23 °C, indicating that a greater proportion of free AA converted to DH-AA during storage at this temperature. This conversion was reflected in the significant decrease in free AA content observed in Figure 5A at 23 °C. In contrast, at 4 °C, the percentage of DH-AA decreased over the storage period (Figure 5C), suggesting that DH-AA is less stable than free AA in strawberry fruit and undergoes degradation over time, even at lower temperatures. Meanwhile, the free AA concentration remained relatively stable at 4 °C (Figure 4). To our knowledge, this study is the first to show that the relative amount of DH-AA increased during storage at 23 °C and decreased at 4 °C in strawberries. These results indicate that cold storage inhibits the conversion of free AA to DH-AA.

### 3.5. The Effect of Storage Durations on Organic Acid Content

Similarly to AA, storage duration had different effects on the citric acid (the principle organic acid) and succinic acid in strawberry fruits at both temperatures (Figure 6). The concentration of citric acid decreased significantly after three days of storage at 23 °C, from 743.7 at D0 to 618.1 at D3, followed by 484.3 mg/100 g FW at D7. In contrast, citric acid concentration remained stable over the two-week storage period at 4 °C. This finding is consistent with previous studies that observed a decrease in citric acid content as strawberries ripen [44]**.**

Succinic acid, a metabolic product of citric acid in ripening fruits [45], decreased in concentration at both storage temperatures, although the decrease occurred at a faster rate at 4 °C. At 23 °C, the succinic acid content significantly decreased from approximately 5.4 mg/100 g FW at D0 to 3.8 mg/100 g FW at D3, followed by 3.4 mg/100 g FW at D7. At 4 °C, the succinic acid contents were much lower, with values of 3.3 mg/100 g FW at D1, 2.7 mg/100 g FW at D7, and 2.0 mg/100 g FW at D14. These results suggest that succinic acid levels decreased over the storage period; however, the loss was partly compensated by the degradation of citric acid into succinic acid at 23 °C.

## 4. Conclusions

The results of the current study demonstrated that refrigerated storage (4 °C) helped preserve the concentrations of key phytochemicals and micronutrients in strawberries, maintaining the nutritional quality over a two-week storage period. In contrast, storage at 23 °C led to significant increases in certain beneficial compounds, such as total anthocyanins and ellagic acid, which may enhance the nutritional value and antioxidant properties of strawberries. However, the higher temperatures also led to significant losses of ascorbic acid and organic acids, both of which play crucial roles in the fruit’s sensory attributes and overall nutritional quality. These findings suggest that, while refrigeration effectively preserves the nutritional profile of strawberries, room temperature storage may improve certain bioactive compounds, though it may also cause nutrient degradation that could negatively impact the overall quality. Therefore, the optimal storage condition for strawberries depends on the desired balance between maintaining overall nutritional quality and enhancing specific bioactive compounds.

## Figures and Tables

**Figure 1 foods-14-00379-f001:**
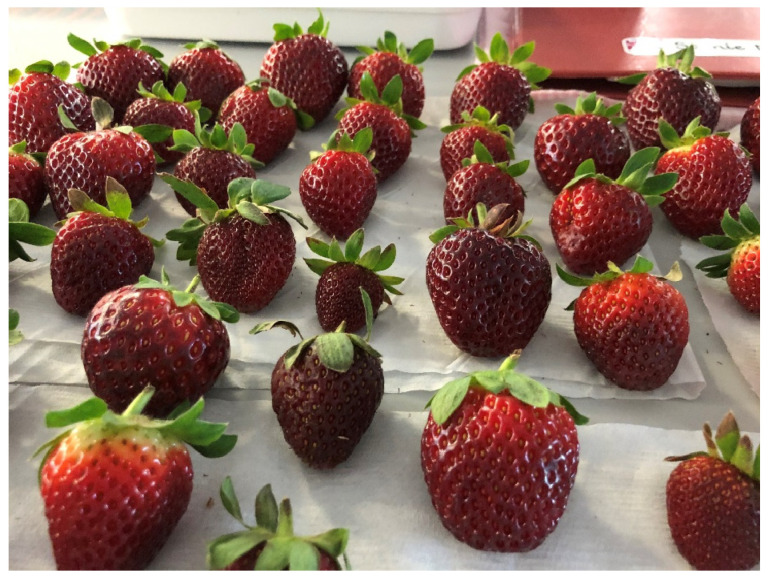
Fresh “Red Rhapsody” strawberries were collected from a commercial farm located in Brisbane, Queensland, Australia.

**Figure 2 foods-14-00379-f002:**
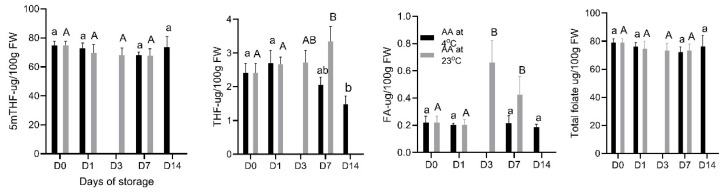
Individual and total folate contents in “Red Rhapsody” strawberries throughout the 14-day storage period at 4 °C and the 7-day storage at 23 °C. Data are presented as mean ± SD (n = 5). Means within columns that are followed by different letters denote significant differences (*p* < 0.05).

**Figure 3 foods-14-00379-f003:**
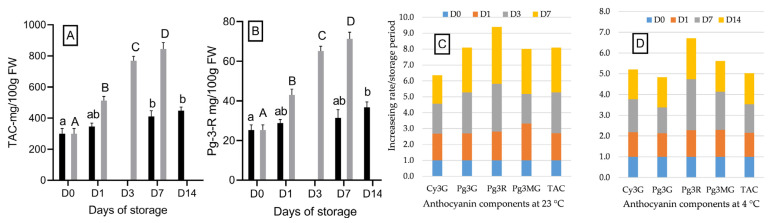
The total anthocyanin (**A**) and pelargonidin-3-rutinoside (Pg-3-R) content (**B**) in “Red Rhapsody” strawberries together with the increasing rates of different anthocyanin components (**C**,**D**) during a 14-day storage period at 4 °C and a 7-day storage period at 23 °C. Data are presented as mean ± SD (n = 5). Means within columns that are followed by different letters denote significant differences (*p* < 0.05).

**Figure 4 foods-14-00379-f004:**
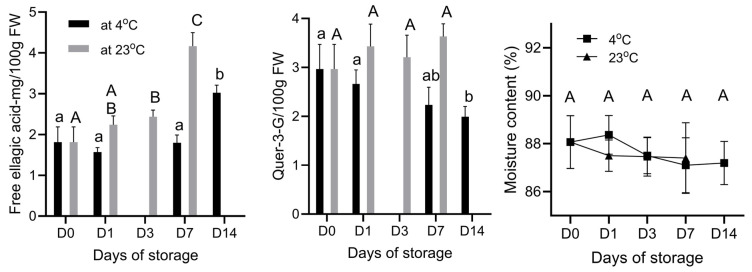
Free ellagic acid, quercetin-3-glucoside, and moisture content in “Red Rhapsody” strawberry fruits during a 14-day storage period at 4 °C and a 7-day storage period at 23 °C. Data are presented as mean ± SD (n = 5). Means within columns that are followed by different letters denote significant differences (*p* < 0.05).

**Figure 5 foods-14-00379-f005:**
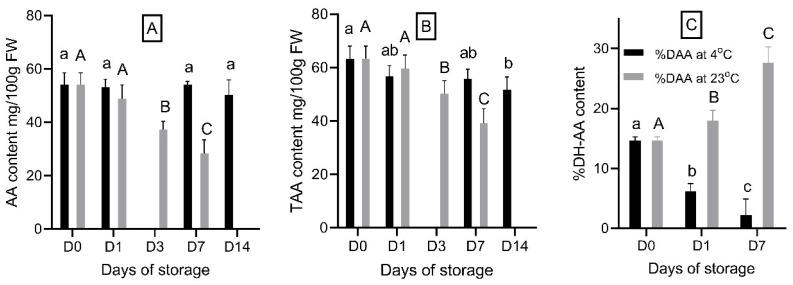
Free AA (**A**), TAA (**B**) content and percentage of DH-AA relative to total AA (**C**) in “Red Rhapsody” strawberry fruits during a 14-day storage period at 4 °C and a 7-day storage period at 23 °C. Data are presented as mean ± SD (n = 5). Means within columns that are followed by different letters denote significant differences (*p* < 0.05).

**Figure 6 foods-14-00379-f006:**
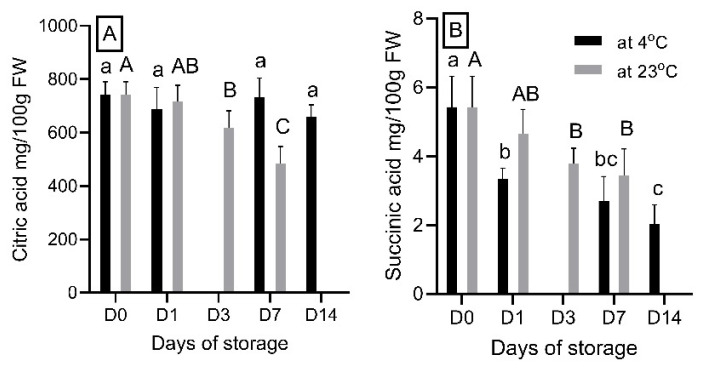
Citric acid (**A**), and succinic acid (**B**) content in “Red Rhapsody” strawberry fruits during a 14-day storage period at 4 °C and a 7-day storage period at 23 °C. Data are presented as mean ± SD (n = 5). Means within columns that are followed by different letters denote significant differences (*p* < 0.05).

**Table 1 foods-14-00379-t001:** MRM scan parameters for citric acid, succinic acid and ascorbic acid (AA).

Compounds	Precursor (*m*/*z*)	Product (*m*/*z*)	Dwell Time (ms)	Q1 Pre Bias (V)	CE (V)	Q3 Pre Bias (V)
Citric acid	191.0	147.2	58.8	11.0	9.0	14.0
	191.0	110.9	58.8	12.0	9.0	14.0
	191.0	103	58.8	14.0	9.0	14.0
	191.0	86.9	58.8	16.0	13.0	19.0
Succinic acid	117.3	99.1	6.0	14.0	13.0	16.0
	117.3	73.05	6.0	12.0	13.0	26.0
AA	175.1	115.2	47.0	24.0	14.0	28.0
	175.1	87.1	47.0	24.0	20.0	28.0

## Data Availability

The original contributions presented in this study are included in the article. Further inquiries can be directed to the corresponding authors.

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
