# Peer review of "Phytochemicals, Organic Acid, and Vitamins in Red Rhapsody Strawberry—Content and Storage Stability"

_foods, 2025, doi:10.3390/foods14030379_

Round 1
Reviewer 1 Report
Comments and Suggestions for Authors
Strawberry is a significant horticultural crop whose fruit are widely consumed for their taste and health benefits. However, the fruit are highly perishable with a short postharvest storage or shelf life. Studying the storage condition on bioactive phytochemicals of the fruit is important for optimizing postharvest handling without compromising their nutritional quality. The present study systematically determined the impact of refrigerated storage (4°C) room temperature storage (23°C) on the contents of anthocyanins, ellagic acid, quercetin-3-glucoside, Ascorbic acid and folate. The findings are of significance to understand how storage conditions, including temperature and duration, impact on the human health-benefitting compounds of fruits. The determinations are accurate and the conclusion is convincing. It is a well-written manuscript.
1. It is generally believed that DH-AA is more stable than free AA. While the present study suggest that DH-AA is less stable than free AA based on their finding that DH-AA decreased during 4°C storge. Can the authors explain the degradation pathway of DH-AA in the present study?
2. Strawberries are highly perishable fruits with fungi diseases, in particular at room temperatures (23°C ) with high humidities (90%). What about the disease index during the storage? The changes in the contents in the fruits with or without disease should be of large difference.
3. The change in the moisture should be included, since the authors used fresh weight.
Minors
1、 In line 309-321, the “free ellagic acid” “total ellagic acid ” should be carefully indicated in the text and the figure 4A.
2、 CAN in line 148 should be in full name.
Author Response
Comments 1: strawberry is a significant horticultural crop whose fruit are widely consumed for their taste and health benefits. However, the fruit are highly perishable with a short postharvest storage or shelf life. Studying the storage condition on bioactive phytochemicals of the fruit is important for optimizing postharvest handling without compromising their nutritional quality. The present study systematically determined the impact of refrigerated storage (4°C) room temperature storage (23°C) on the contents of anthocyanins, ellagic acid, quercetin-3-glucoside, Ascorbic acid and folate. The findings are of significance to understand how storage conditions, including temperature and duration, impact on the human health-benefitting compounds of fruits. The determinations are accurate and the conclusion is convincing. It is a well-written manuscript.
Response 1: thank you for the encouraging comments. We appreciate your positive feedback on our study.
Comments 2: it is generally believed that DH-AA is more stable than free AA. While the present study suggests that DH-AA is less stable than free AA based on their finding that DH-AA decreased during 4°C storge. Can the authors explain the degradation pathway of DH-AA in the present study?
Response 2: thank you to Reviewer 1 for the insightful comment. We agree that DH-AA is generally considered more stable than free AA, and we are also aware of the interconversion between DH-AA and AA in fruits. In our study, we believe that the storage at 23°C is closer to the natural growing temperature of strawberries, so the fruit stored at this temperature behaves similarly to its conditions in the field. In contrast, 4°C is too cold for the fruit, activating its natural resistance mechanisms. This likely leads to an increase in antioxidant compounds through the interconversion of DH-AA to AA. As a result, while the total ascorbic acid (TAA) level remained stable, DH-AA levels significantly decreased during storage at 4°C.
Comments 3: strawberries are highly perishable fruits with fungi diseases, in particular at room temperatures (23°C) with high humidities (90%). What about the disease index during the storage? The changes in the contents in the fruits with or without disease should be of large difference.
Response 3: thank you to Reviewer 1 for the insightful suggestion. We appreciate your input. During our trials, we did observe fungal diseases affecting the strawberries. However, the primary focus of this study was to evaluate the changes in phytochemicals, organic acids, and vitamins in the edible parts of the fruit. We did not specifically collect data on disease progression, but we agree that this would be an interesting area for future research.
Comments 4: the change in the moisture should be included, since the authors used fresh weight.
Response 4: We have revised the text as follows: A new paragraph has been added to the manuscript: The moisture content of strawberry fruit at both storage temperatures was not statistically significant (p > 0.05). These results suggest that the storage conditions effectively maintained the moisture content in the strawberries. Given the stability of moisture content, the observed changes in phytochemicals, organic acids, and vitamins in the strawberries reflect the physiological processes occurring in the fruit during storage, rather than being influenced by moisture changes. Figure 4 has been updated
Comments 5: In lines 309-321, the “free ellagic acid” “total ellagic acid” should be carefully indicated in the text and the figure 4A.
Response 5: thank you to Reviewer 1, revised as follows: free ellagic acid has been added to the heading 3.3. and figure 4A to indicate that our study focuses on free form of ellagic acids rather than total ellagic acid.
Comments 6: CAN in line 148 should be in full name.
Response 6: revised as follows: ACN was replaced by acetonitrile
Reviewer 2 Report
Comments and Suggestions for Authors
General comment:
The citation format of the entire manuscript does not meet editorial requirements and must be revised.
Abstract
I suggest removing or explaining the abbreviation FW because it is not commonly used or known.
Material and methods
Why were storage tests conducted at 90% relative humidity? Such information should be included in the methodology (preliminary studies) or in the Introduction if literature reports were used.
Line 57 and 90: In my opinion capital letters in “Vitamin’ and ‘Ascorbic’ are not necessary. This also applies to the rest of the manuscript.
Line 111 and 213: missing hygrometer and shaker specifications, manufacturer, city, country.
Results
Figures 2-6 need to be improved in quality and have a clear legend on them.
Line 260: What kind of processes caused the increase in THF content on Day 7? Literature reference or deeper discussion required.
Did the Authors take into account the change in fruit weight during storage? The methodology mentions determination of dry weight but no results were presented. Therefore, could the increase in some bioactive compounds during storage not be due to the fact that the dry mass of the material increased as a result of moisture loss? Why was it decided to express the content of components in terms on fresh weight?
Author Response
Abstract
Comments 1: I suggest removing or explaining the abbreviation FW because it is not commonly used or known.
Response 1: Thank you for your suggestion. The abbreviation "FW" has now been explained in the abstract (line 22) for better clarity.
Material and methods
Comments 2: Why were storage tests conducted at 90% relative humidity? Such information should be included in the methodology (preliminary studies) or in the Introduction if literature reports were used.
Response 2: Thank you for your comment. The storage tests were conducted at 90% relative humidity because the moisture content in strawberries is approximately 90%. Additionally, a previous study (Hong, Phan, & O'Hare, 2021) also used 90% relative humidity for storing sweetcorn. To clarify this, we have added the following sentence to the methodology: "The humidity level was set based on the moisture content of strawberry fruit and previous publications (Hong et al., 2021)
Comments 3: Lines 57 and 90: In my opinion capital letters in “Vitamin’ and ‘Ascorbic’ are not necessary. This also applies to the rest of the manuscript.
Response 3: Thank you for your comment. The capital letters in “Vitamin” and “Ascorbic” have been corrected and replaced with lowercase throughout the manuscript, as suggested.
Comments 4: Line 111 and 213: missing hygrometer and shaker specifications, manufacturer, city, country.
Response 4: Thank you for your comment. The manufacturer, city, and country for both the hygrometer and shaker have been added to the text for clarity.
Comments 5: Figures 2-6 need to be improved in quality and have a clear legend on them.
Response 5: Figures 2-6 have been replaced by better resolution (600dpi) and their legends have been updated
Comments 6: Line 260: What kind of processes caused the increase in THF content on Day 7? Literature reference or deeper discussion required.
Response 5: Thank you for your comment. Two references have been added to the text to support the explanation regarding the interconversion of folate components, which may have contributed to the increase in THF content on Day 7. While we are uncertain about the exact cause of the increase in THF content, we hypothesize that after 7 days of storage at room temperature, the fruit's texture may have been damaged, and the enzyme responsible for converting THF to 5mTHF may no longer have been active, leading to an increase in THF levels.
Comments 8: Did the Authors take into account the change in fruit weight during storage? The methodology mentions determination of dry weight but no results were presented. Therefore, could the increase in some bioactive compounds during storage not be due to the fact that the dry mass of the material increased as a result of moisture loss? Why was it decided to express the content of components in terms on fresh weight?
Response 8: Thank you for your comment. We have revised the text as follows: A new paragraph has been added to the manuscript: "The moisture content of strawberry fruit at both storage temperatures did not change significantly (p > 0.05). These results suggest that the storage conditions effectively maintained the moisture content in the strawberries. Given the stability of moisture content, the observed changes in phytochemicals, organic acids, and vitamins reflect the physiological processes occurring in the fruit during storage, rather than being influenced by moisture changes." Additionally, Figure 4 has been updated.
Regarding your question about fruit weight, we expressed the content of components in terms of fresh weight, as strawberries are typically consumed fresh, and this approach better reflects the conditions under which they are commonly used. While dry weight determination could provide insights into moisture loss, the stable moisture content observed in this study minimizes its impact on the changes in bioactive compounds.
Reviewer 3 Report
Comments and Suggestions for Authors
The study of storage conditions is simplified to the home preservation of fresh fruits. This is an unrealistic scenario, as fruits typically do not come from a single harvest of fruits grown in the same field. The statistical approach in evaluating the variation of the considered parameters is weak as long as only the variance of the data is assessed without considering measurement uncertainty. The instrumental and analytical resources used are considerable, possibly redundant when contextualized relative to the type of study conducted. Multivariate statistical analysis could have provided information to deepen various aspects of the compositional variation due to the fruit's storage conditions.
Despite this, the study was carried out in alignment with the set objective, gathering useful information even though it pertains more to the experimental setup than to real-world conditions. While the experiment was conducted under controlled circumstances, the findings still provide valuable insights, especially regarding the factors affecting preservation and fruit composition. It is important to recognize that experimental setups often simplify real-world complexities, but the results still contribute to understanding the underlying principles that could be applicable under different conditions.

Author Response
Comments 1: The study of storage conditions is simplified to the home preservation of fresh fruits. This is an unrealistic scenario, as fruits typically do not come from a single harvest of fruits grown in the same field. The statistical approach in evaluating the variation of the considered parameters is weak as long as only the variance of the data is assessed without considering measurement uncertainty. The instrumental and analytical resources used are considerable, possibly redundant when contextualized relative to the type of study conducted. Multivariate statistical analysis could have provided information to deepen various aspects of the compositional variation due to the fruit's storage conditions.
Despite this, the study was carried out in alignment with the set objective, gathering useful information even though it pertains more to the experimental setup than to real-world conditions. While the experiment was conducted under controlled circumstances, the findings still provide valuable insights, especially regarding the factors affecting preservation and fruit composition. It is important to recognize that experimental setups often simplify real-world complexities, but the results still contribute to understanding the underlying principles that could be applicable under different conditions.
Response 1: Thank you for your thoughtful comments. We appreciate your feedback on our manuscript. We acknowledge that the study of storage conditions was simplified to focus on home preservation of fresh fruits, which may not fully reflect real-world scenarios, as fruits typically come from multiple harvests and different fields. Regarding the statistical approach, we understand your concern about the variation analysis, as only the data variance was considered without accounting for measurement uncertainty. We agree that incorporating such uncertainty could improve the robustness of the analysis.
We also appreciate your suggestion about the use of multivariate statistical analysis, which could have offered deeper insights into the compositional variation due to storage conditions. While we used multiple instrumental and analytical resources, we recognize that some of these may seem redundant relative to the scope of the study. However, we believe they were necessary to achieve the level of precision required for this type of research.
Despite these limitations, the study was conducted in alignment with the set objectives, providing valuable information. Although the experimental setup may not fully reflect real-world conditions, the findings still offer meaningful insights into the factors influencing preservation and fruit composition. The results contribute to a better understanding of the principles that could be applied in various conditions.
Comments 2: Line 177: “Chromatographic separation for AA and other organic acids was carried out on an UPLC BEH AcclaimTM C30 column (250 x 2.1 mm i.d., 3.0μm particle size; Thermo Scientific, Waltham, MA USA),”
From what is stated, it seems that the AA concentration was determined using two different instruments and methods, which appears redundant since both involve triple quadrupoles. It would be better to avoid mentioning both methods if the results are not meaningfully discussed or compared.
Response 2: Thank you for your thoughtful comments. I believe there may have been a misunderstanding. The UPLC column from Thermo Scientific was used for the separation of organic acids, while the Shimadzu LC-MS/MS system was employed for the identification and quantification of ascorbic acid (AA) and other organic acids. To avoid confusion for future readers, the sentence has now been revised to: "Chromatographic separation of AA and other organic acids was carried out on a UPLC column (BEH AcclaimTM C30, 250 x 2.1 mm i.d., 3.0μm particle size; Thermo Scientific, Waltham, MA, USA)."
Comments 3: Line 219: The moisture content parameter was neither used nor sufficiently discussed to evaluate storage conditions. It might be a useful factor, but its relevance remains unclear due to the lack of explanation.
Response 3: We have revised the text as follows: A new paragraph has been added to the manuscript: The moisture content of strawberry fruit at both storage temperatures was not statistically significant (p > 0.05). These results suggest that the storage conditions effectively maintained the moisture content in the strawberries. Given the stability of moisture content, the observed changes in phytochemicals, organic acids, and vitamins in the strawberries reflect the physiological processes occurring in the fruit during storage, rather than being influenced by moisture changes. Figure 4 has been updated
Comments 4: Line 230: Without accounting for measurement uncertainty, ANOVA can lead to misleading conclusions, as observed variations might stem from measurement errors rather than true effects.
Response 4: Thank you for your valuable feedback. We agree that without accounting for measurement uncertainty, ANOVA may lead to misleading conclusions, as observed variations could be attributed to measurement errors rather than actual effects. In future studies, we will consider incorporating measurement uncertainty into the analysis to improve the reliability of our studies. This approach would provide a more accurate assessment of the true effects and help ensure the robustness of the statistical findings.
Comments 5: Line 280: “Although the anthocyanin profile remained unchanged,”
Actually, according my opinion, the profile, understood as the percentage abundance of individual compounds, appears to vary when considering the increase rate, as well as the ratio between the initial concentration and the concentration after prolonged storage.
Response 5: Thank you for your valuable feedback. Line 280 now is written “Although the anthocyanin components remained unchanged”
Comments 6: Line 305: The increasing rate was calculated using an equation that was not provided or explained
Response 6: Revised as follows: Figure 5. Free AA (A), TAA (B) content and percentage of DH-AA relative to total AA (C) in ‘Red Rhapsody’ strawberry fruits during a 14-day storage at 4 °C and 7-day storage at 23 °C.
Comments 7: Line 388: Without accounting for measurement uncertainty, ANOVA can lead to misleading conclusions, as observed variations might stem from measurement errors rather than true effects. A multivariate data analysis could be more useful in this context, as it highlights which factors influence measurement variations, considering interactions between variables and the associated uncertainty.
Response 7: Thank you for your insightful comment. We agree that measurement uncertainty is an important consideration in data analysis and can potentially influence the interpretation of results. In our study, we performed a one-way ANOVA to assess the main effects of the storage conditions on the measured parameters. However, we acknowledge that this approach does not fully account for measurement uncertainty and interactions between variables.
We appreciate your suggestion to employ multivariate analysis, which would indeed allow for a more comprehensive understanding of the factors influencing variation in the data, as well as the uncertainty associated with these measurements. In light of your comment, we will consider incorporating multivariate statistical techniques, such as Principal Component Analysis (PCA) or analysis of covariance (ANCOVA), in future work to better capture the interactions between storage conditions and other factors. This approach would provide a more robust analysis by considering the variability in the data and the associated measurement uncertainty.
We value your input and will take it into account to improve the robustness of our future analyses.